# Bacillus Calmette–Guérin Immunotherapy for Cancer

**DOI:** 10.3390/vaccines9050439

**Published:** 2021-05-01

**Authors:** Fabíola Cardillo, Maiara Bonfim, Periela da Silva Vasconcelos Sousa, José Mengel, Luiz Roberto Ribeiro Castello-Branco, Rosa Teixeira Pinho

**Affiliations:** 1Laboratory of Molecular and Structural Pathology, Gonçalo Moniz Institute, FIOCRUZ, Salvador, BA 40296-710, Brazil; maiara.bonfim@ufba.br; 2Laboratory of Clinical Immunology, Oswaldo Cruz Institute, FIOCRUZ, Rio de Janeiro, RJ 21040-900, Brazil; perielavasconcelos@id.uff.br (P.d.S.V.S.); jomengel@ioc.fiocruz.br (J.M.); rospinho@ioc.fiocruz.br (R.T.P.); 3Laboratory of Molecular Virology and Marine Biotechnology, Fluminense Federal University, Niteroi, RJ 24220-008, Brazil; 4Faculty of Medicine of Petropolis, UNIFASE, Petropolis, RJ 25680-120, Brazil; 5Ataulpho de Paiva Foundation, Rio de Janeiro, RJ 20941-070, Brazil; branco@bcgfap.com.br

**Keywords:** BCG, cancer, trained immunity

## Abstract

Bacillus Calmette–Guérin (BCG), an attenuated vaccine from *Mycobacterium bovis*, was initially developed as an agent for vaccination against tuberculosis. BCG proved to be the first successful immunotherapy against established human bladder cancer and other neoplasms. The use of BCG has been shown to induce a long-lasting antitumor response over all other forms of treatment against intermediate, non-invasive muscle bladder cancer Several types of tumors may now be treated by releasing the immune response through the blockade of checkpoint inhibitory molecules, such as CTLA-4 and PD-1. In addition, Toll-Like Receptor (TLR) agonists and BCG are used to potentiate the immune response against tumors. Studies concerning TLR-ligands combined with BCG to treat melanoma have demonstrated efficacy in treating mice and patients This review addresses several interventions using BCG on neoplasms, such as Leukemia, Bladder Cancer, Lung Cancer, and Melanoma, describing treatments and antitumor responses promoted by this attenuated bacillus. Of essential importance, BCG is described recently to participate in an adequate microbiome, establishing an effective response during cell-target therapy when combined with anti-PD-1 antibody, which stimulates T cell responses against the melanoma. Finally, trained immunity is discussed, and reprogramming events to shape innate immune responses are addressed.

## 1. Introduction

Cancer is a leading cause of death, with about 9.6 million deaths and 18 million new cases worldwide [1]. The study of Bacillus Calmette–Guérin (BCG) anti-tumor activity began in 1929 when Pearl (1929) [2] observed a lower frequency of cancer in necroscopy in patients with tuberculosis at Johns Hopkins Hospital. Thus, from 1959 onwards, the use of BCG in oncology was proposed based on experimental studies in mice that showed more (or higher) resistance to tumor implantation in animals treated with BCG [3] and several studies on the anti-tumor effect of BCG have been developed [4,5,6,7,8].

BCG is used as a vaccine obtained from an attenuated strain of *Mycobacterium bovis*, developed from an initially more virulent strain, and described by Calmette and Guérin (1908, Pasteur Institute) [9]. All BCG preparations worldwide are derived from this original strain [10]. Developed initially as an attenuated agent for vaccination against tuberculosis, BCG has been described as responsible for the non-specific increase in the immune system’s activity responding to a variety of neoplasms, causing them to regress at experimental levels [11]. BCG was initially described as an agent that produces a potent intra-tumoral cellular inflammatory response that somehow would induce tumor-shrinking.

## 2. Bladder Cancer

Bladder cancer (BC) is the second most common type of urinary tract cancer, ranking fourth (10% of cases) in men and eighth place (4% of cases) in women [12]. Although there are familial case reports, it is known that BC’s occurrence is more related to exogenous factors than to genetic factors. Among the risk factors, smoking is the most significant, with about 50% males and 35% females [12]. In addition, environmental exposure to tobacco smoke increases urothelial developing BC’s risk [13]. Tobacco contains aromatic amines and polycyclic aromatic hydrocarbons, excreted by the kidneys [14].

With about 10% of all cases, the second most significant risk factor is occupational exposure to polycyclic aromatic hydrocarbons, chlorinated hydrocarbons, and aromatic amines. Exposure is mainly due to industrial causes for processing paints, dyes, metals, and petroleum derivatives. Thus, to minimize the risks, occupational safety guidelines have been implemented in developed industrial environments [14].

The BCG vaccine’s anti-tumor effect hypothesis emerged in the early twentieth century when researcher Pearl (1929) [2] observed a lower frequency of cancer in necroscopy of tuberculosis patients at Johns Hopkins Hospital. BCG cancer therapy began to be used and studied in 1930, and from this, its antitumor effect was demonstrated in clinical and laboratory trials in malignant cell lines [4,5,6,7]. Morales and colleagues (1976) [8] were the first to report BCG’s effectiveness as an adjunct therapy in superficial bladder cancer. They conducted the first clinical trial administering intravesical BCG to patients with urothelial bladder cancer once a week for six weeks. The treatment was not only the first successful immunotherapy confirmed against an established solid human cancer, but it proved to be lasting over all other forms of medical treatments for intermediate and high-risk non-invasive muscle bladder cancer [15]. The results were impressive, showing a decrease in the recurrence rates of the disease. Then, with more advanced studies from the Southwestern Oncology Group (SWOG) and the Memorial Sloan–Kettering, Morales’s proposed scheme of intravesical BCG therapy for muscle-invasive urothelial BC came to be accepted and applied [16,17]. Studies to evaluate other BCG strains, including the Moreau-Rio de Janeiro, in patients with superficial bladder cancer stratified by risk groups gave similar results to different strains worldwide [18,19]. The fact that BCG is a living organism and its antineoplastic activity mechanisms are immunologically mediated addresses challenges to understand how BCG works in these systems [20,21,22,23,24].

Furthermore, specific T cell activity against tumor cells has been studied [22,23]. After BCG intravesical treatment on an experimental study of bladder cancer, the authors detected a higher number of CD4 T-cells infiltrating the tumor (TIL) and correlated it with increased IFN-γ by these specific CD4 T cells. They suggested that the BCG-treated mice increased longevity was related to tumor-intrinsic IFN-γ signaling, since class II-MHC was not necessary to be present in the infiltrated tumor [23].

Non-invasive muscular bladder cancer (NMIBC) comprises cancer affecting the mucosa classified as Tumor categories (T) and in very early stages are known as Ta (non-invasive papillary carcinoma) and Tis (carcinoma in situ) [25,26]. In these early stages, transurethral resection (TUR) can easily remove the tumor. An invasive submucosa form is classified as the T1 stage, where cancer spreads to the connective tissue without involving bladder wall muscle [26,27]. Despite the increased incidence of bladder tumors in recent decades, survival has improved with advances in therapy. The treatment of superficial bladder cancer considers that the gold standard is TUR. However, there is a risk of recurrence (60–90%) and disease progression (30%) [28]. Thus, intravesical chemotherapy and BCG immunotherapy are indicated. BCG instillations are currently considered standard therapy for non-invasive high and medium-risk muscle bladder cancer (NMIBC) [26].

Recently, a recombinant diadenylate synthase gene (disA)-overexpressing BCG strain called BCG-disA-OE was developed [29]. The disA gene is expressed at low levels in wild-type BCG strains. The overexpression of the diadenylate synthase gene allows the microbe to release large amounts of the STING (stimulator of interferon genes) agonist c-di-AMP1. STING is part of the cytosolic surveillance pathway (CSP) and responds to DNA and cyclic dinucleotides aberrantly present in the cytosol [29,30,31,32,33,34]. STING stimulation activates the production of type I interferon and NF-kB-dependent immune responses, increasing dendritic T cell-priming and the recruitment of effector T cells [35]. Preclinical studies show an augmented antitumor activity of the recombinant BCG compared with the wild-type BCG in experimental bladder cancer models [35].

Although BCG is considered the standard therapy for individuals with NMIBC, it has a counterpoint to the high recurrence rates and, in some cases, the bladder removal due to the lack of response. A recent study showed that patients with NMIBC revealed an impressive improvement in BCG response when immunized percutaneously before intravesical γδ T and NK cell’s weak cytotoxic responses were reported and associated with poor local BCG responses, a situation reversed by previous systemic immunization [36]. Therefore, the search continues for advances in BCG immunotherapy and mechanisms of action [37].

## 3. Leukemia

Leukemia refers to malignant disorders characterized by the increased number of leukocytes either in the blood or bone marrow. The types of leukemia are called basically according to the affected cells. Chronic lymphocytic leukemia (CLL), for example, may account for mature cells, acute leukemia, a precursor of various cell-lineages, whereas chronic myeloid leukemia (CML) cells may be mature [37]. Leukemia cancer incidence reached ninth in the ranking of deaths in 2015 and ranked eighth in cancer incidence globally in the same year. In 2015 there were 606,000 new cases of leukemia worldwide and 353,000 deaths. One in 87 men than 1 in 137 women developed leukemia between ages 0 and 79 globally [38]. According to the Brazilian National Cancer Institute (INCA, Rio de Janeiro, Brazil) (2018) [39], there were 10,800 new cases of leukemia in 2018, 2.8% in men and 2.4% in women, in Brazil. In children and adolescents, leukemia is the most common cancer [39].

There are several risk factors for developing leukemia in genetically predisposed individuals, such as paternal smoking, ionizing radiation during prenatal and postnatal life, exposure to domestic pesticides and benzene, all potentially carcinogenic [40]. Various infections are related to cancer development during childhood, and vaccination is indicated to avoid an increase in cancer cases [41]. Some hematological diseases are related to a viral origin, and the retrovirus human T-cell lymphotropic virus 1 (HTLV1) was a causative factor for adult T-cell leukemias and lymphoma. Thus, childhood leukemia might have an infectious, possibly a retroviral cause. Greaves describes that between the hypotheses studied, the most common is that one or more infections are acquired during childhood socio-demographic circumstances, causing leukemia [41].

Thus, recently, Morra et al. (2017) [42], in a meta-analysis study, sought to evaluate the association between vaccination and childhood leukemia. The vaccines studied were BCG vaccine, hepatitis B vaccine (HBV), *Haemophilus influenza* type B (HiB) and measles, rubella, mumps (MMR) trivalent vaccine. Among the vaccines analyzed, they observed a protective association of BCG vaccine in the first year of life and childhood leukemia risk. These results attract a motivating factor for the development of further studies of the anti-tumor effect of BCG.

## 4. Lung Cancer

Lung cancer (both small cell-SCLC and non-small cell-NSCLC) is the second most common cancer [43] and is among the most frequently diagnosed cancers. However, it has low overall survival and is responsible for the highest number of deaths due to poor prognosis and difficult early detection [44]. Smoking is the major risk factor for lung cancer development resulting in approximately 80% of deaths [43]. The association between chronic inflammation and the increased risk of cancer development is well described [45,46]. Although about 20% of cancers are related to chronic inflammation, innate immune cells and mediators are found in most human neoplasms [45,47]. This inflammatory microenvironment comprises tumor-infiltrating inflammatory cells, tumor-associated fibroblasts, as well as endothelial progenitor cells [46,48]. Tumor and host cells produce cytokines and chemokines in this microenvironment, allowing the coordination of a self-limiting immune response [49]. It was then hypothesized that the immunomodulatory effect of BCG is a stimulating alternative microenvironment. Azuma et al., (1971) [50] in the 1970s identified and isolated a bioactive component of the BCG cell wall called the BCG cell wall skeleton (BCG-CWS). However, these results aroused more significant interest in the scientific community to understand the BCG-CWS components responsible for its adjuvant cancer action. In the late 1970s, Yasumoto et al. (1979) [51] and others [52] observed increased survival of BCG-CWS-treated subjects compared to the control group in a clinical study on lung cancer or of BCG on melanoma patients [24]. Years later, the clinical application of BCG-CWS began in immunotherapy for lung and gastric cancer [52,53].

It was also described that BCG has critical components to stimulate the immune response, such as mycolic acid, arabinogalactan, peptidoglycan, and these substances are natural ligands for TLR2 and TLR4 receptors (TLRs) [54]. Recognition of these BCG wall components by pattern recognition receptors (RRPs), TLRs, and C-type lectin receptors (CLRs) stimulate differentiation of dendritic cells (DCs) into antigen-presenting cells (APCs) [55]. Cytokine and protein production occurs by these cells and may induce inflammation and an adaptive cellular immune response [55]. Therefore, preclinical cancer vaccine studies using BCG-CWS have shown their practical immune adjuvant effect by inducing tumor-specific T cells to a degree sufficient to eradicate established mouse tumors [56,57]. Among these vaccines is the peptide vaccine using a tumor-associated antigen peptide (TAA). Many TAAs have been identified and used as therapeutic cancer vaccines [57,58]. The most promising TAA is the Wilms tumor gene product (WT1), which is highly expressed during acute leukemia and found in various hematopoietic types of neoplasms. Therefore, it is a useful marker for targeted immunotherapy [59]. Recently, Nishida et al., (2019) [60] performed a phase I dose-escalation study of BCG-CWS in association with WT1 peptide in patients with advanced cancer. They observed good tolerance to the administered dose of BCG-CWS and clinical effects in several patients bearing advanced cancer, including non-small cell lung cancer (NSCLC) and melanoma. Differentiation of naive CD4^+^ T cell-subset to the memory phenotype was observed in some patients, resulting in an antitumor response. The immunomodulatory activity of BCG-CWS and consistent antitumor response may suggest a blockade of tumor immunosuppressive microenvironment.

## 5. Melanoma

Since the 1970s, melanoma is considered the leading cause of death from skin cancer and has increased [1]. It is the most aggressive among dermatological cancers, representing 4% [61]. The invasive behavior of breast cancer and melanoma cells seems to be reminiscent of their origin in the neural crest [62]. Its development is related to several genetic and environmental factors, such as excessive exposure to ultraviolet radiation associated with its high metastatic potential [63].

The diagnosis of melanoma is made by dermoscopic examination and advanced computer digital imaging techniques, followed by confirmation by biopsy and histopathological examination to classify the staging. Approximately 50% of melanoma cases have a mutation in the BRAF V600E gene, contributing to tumor growth, angiogenesis, and metastatic progression [63,64]. The mutation resulting from the V600 K is the most observed [65], leading to activation of the MAPK pathway, which regulates average growth and survival cells [66,67,68]. Thus, these mutations and epigenetic changes promote and stimulate the secretion of growth factors that contribute to cell proliferation, angiogenesis, changes in the extracellular matrix, and metastasis [69].

Since melanoma has a high probability of inducing metastases and spreading to other organs, several therapeutic trials are being followed [70,71]. Treatment for metastatic melanoma (MM) is based on systemic therapy, radiation therapy, and surgery. However, no increase in patient’s survival with these treatments was observed, causing adverse effects [72]. Thus, the search for new therapies to reduce the metastatic potential has created space in the scientific community. Since 2011, the FDA (USA) has approved ten new candidates, including targeted therapies, immunotherapies, and cancer vaccines [71,73]. Although advances have been made, these therapies still have certain limitations, such as low response rates, side effects, and resistance [74].

Other immune effectors, such as macrophages, and T cells, have been suggested to participate in BCG immune response [20]. In studies published in 2017 by Lardone and collaborators [20], BCG has been shown to alter the melanoma microenvironment by favoring a T cell response against the tumor. For this reason, it is essential to study the characteristics of the immune response mediated using BCG in melanoma.

As previous results in the use of BCG in melanoma, in studies by Morton (1974) [24], about 90% of the malignant nodules regressed in immunologically competent patients. Occasionally, the malignant nodules found in distant places where BCG was inoculated also regressed in some patients [29]. In this study, a patient (meaning 40%) remained completely tumor-free for two years. It was concluded that patients who received the BCG vaccine might have a lower recurrence rate and a higher survival rate than those treated elsewhere, without the vaccine. However, some of the potential dangers of this type of therapy can be emphasized [24]. BCG injections can result in fevers, chills, and abscesses at the injection sites [75]. These side effects have been reported after repeated administration of large doses of the vaccine [76]. Lowering this vaccine’s amount can bring beneficial results to the patient and should still be tested [28].

The study of cellular populations involved in antimelanoma response and functional activities is fundamental to elucidate the mechanisms involved in new satisfactory immunotherapy. Adjuvant therapies together, such as the imiquimod (IMIQ) reagent, a TLR-7/8 agonist associated with BCG, are being tested, evaluating the efficacy and ability of combined treatments to modulate the immune system response better [75]. Preclinical studies employing the combination of IMIQ with a *Listeria monocytogenes*-based vaccine have shown a drastic increase in local and systemic antimelanoma immunity [77]. In these preclinical studies, both the Listeria vaccine and the IMIQ individually offered partial control of local tumors and pulmonary metastases. Still, the combination led to a profound rejection of the tumor in a highly reproducible manner [75,77].

In humans, the common side effects of the combination of BCG and IMIQ were mild pain, in the site reaction (moderate erythema, induration, and ulceration), and moderate fevers [75]. Inflammatory responses to IMIQ induced similar local toxicity. There were no adverse events related to treatment, and there were no cases of systemic BCG infection. Because the results for this small series of stage III patients are surprisingly favorable, additional clinical and laboratory evaluations are needed to determine whether this combination induces systemic immunity. Regional control of the tumor with this combination was excellent, and with selective surgical resection, 78% of the patients achieved complete and durable control of the disease. In addition, no patient in this series of studies died of melanoma.

Kidner et al. in 2013 [75] reported the combined use of IMIQ with intralesional inoculated BCG (IL-BCG) in nine patients with stage III melanoma. Three of the patients underwent resection of resistant solitary lesions. These resections were performed during the regression of other disease sites, and the three patients had complete resolution of melanoma. Until the last follow-up, six patients remained without evidence of disease. Following studies, one patient developed additional lesions in transit after seven months and then had to receive different local immunotherapy with a combination of IL-BCG and IMIQ. Another patient recalled the disease in transit after 34 months, and another developed pulmonary metastasis after 12 months. Two patients died of non-melanoma-related causes after 17- and 55-months post-treatment.

Since all treatments commonly used in advanced melanoma cases, such as chemotherapy, radiotherapy, and vaccines, have resulted in a few cure cases, early diagnosis remains the main objective in managing patients with melanoma, aiming to cure [78]. Although recent efforts try to bring more effective therapies in metastatic melanomas (or not), promising treatments are in the final clinical trials phase [77]. This area’s results have been attributed to two main approaches: immunotherapy and target therapy as fundamental to understanding the cellular mechanisms involved in these approaches through experiments that indicate a better prognosis for melanoma patients.

The specific blocking of checkpoints during immunological therapies has been used for the treatment of advanced melanoma. Target tools that inhibit target molecules have been developed, such as the 04-Antigen associated with cytotoxic T lymphocytes (CTLA-4) and the programmed death axis PD-1/PD-L1 [79]. As CTLA-4 inhibitors, the anti-CTLA-4 antibody inhibits CTLA-4-binding of T cells to the CD80 (B7.1) and CD86 (B7.2) APC ligands. This intervention can restore T cell proliferation and develop an effective immune response to Tumor-Associated Antigens (TAAs) [80]. The treatment with Anti-CTLA-4 antibodies (Ipilimumab) was the first among treatments with immunological blockers to inhibit the action of CTLA-4 in the immune tolerance to tumor cells. This treatment was approved in 2011 by the FDA in four doses of 03 mg/kg every three weeks in patients with stage III metastatic melanoma tumors [79,80].

It has also been previously reviewed that several inhibitors of responses against the tumor promote inactivation or cell death during the immune response [81]. Regarding PD-1 inhibitors, the Anti-PD-1 antibody has the function of binding to the PD-1 receptor in activated T cells and inhibiting the PD-1/PD-L1 interaction, stimulating T cell’s response against the tumor. In 2014, the FDA approved two treatments with Anti-PD-1 (pembrolizumab and nivolumab) to treat patients with advanced melanoma [71]. PD-L1 inhibitors are antibodies that also block the PD-1/PD-L1 axis. PD-L1 is expressed in tumor cells, inhibiting T cell activation by recognizing and interacting with PD-1 of them [71].

Tumors, such as melanoma, express high levels of PD-L1 correlated with lymphocytes infiltrated into the tumor. There are three treatments related to PD-L1 inhibitors (atezolizumab, avelumab, and durvalumab), which are currently approved for treatments of several malignant neoplasms. Clinical research has suggested that response rates of patients with melanoma treated with Ipilimumab alone are 15%. However, with Ipilimumab combined with an anti-PD-1 therapy was improved to 35–40% [71,80,82]. Checkpoint inhibitors may not induce effective antitumor immune responses in the majority of patients [83]. A better antitumor activity may be related to cancer neoantigen’s appearance generated by tumor gene mutations during the treatment [84,85]. However, factors beyond genomics, such as the microbiome, are involved in the immune response to developing melanoma [86,87]. For instance, studies demonstrate that the gut microbiota modulates the response to anti-PD-L1 in patients, and an appropriate microbiome can assist these blocking therapies [86,87] Furthermore, germ-free mice were reconstituted with fecal material from patients that responded positively to a checkpoint blockade therapy [86]. This treatment improved tumor control, causing increasing T cell responses and greater efficacy of anti-PD-L1 therapy [86]. Therefore, the presence of specific microbes in the microbiome would favor the activation of cross-reactive T cell effectors that could help control tumor growth. Moreover, diet components could act as immune modulators by providing cross-reactive peptides between food proteins and self-antigens, influencing the T cell repertoire composition and activation upon checkpoint inhibitor treatment [88,89]. Of course, peptide cross-reactivity would be dependent on the Human leukocyte antigens/Major Histocompatibilty (HLA/MHC) haplotype association [90,91]. The recent demonstration of microorganisms from the microbiome, infecting tumor cells and microorganism-derived peptides being presented to T cells by tumor cells themselves, opens up another interesting hypothesis to explain the differential effects of checkpoint inhibitors among patients with different microbiomes [92].

New strategies that enhance therapies against metastatic melanoma (MM) have been studied using BCG.As mentioned earlier, BCG’s clinical application in immunotherapy for melanoma started in the 1970s with BCG-CWS, a bioactive component of the cell wall that is a robust immunological adjuvant for immunotherapy [60].

BCG is currently used as an immunotherapeutic by intralesional injection to treat MM at various stages and can be administered alone or associated with a vaccine or autologous tumor cell medication [61]. Exciting results have been observed and included other cancers, such as melanoma and stomach cancer, for local and regional tumor regression with improved patient survival [57]. A study of the association of BCG with Canvaxin™, an allogeneic vaccine, after complete resection in patients with stage IV melanoma, resulted in antitumor immune responses and an overall survival rate of 39% over five years [93]. In 2017, Faries et al. [94] performed this same study model in 246 patients with stage IV melanoma. A 5-year survival rate of 39.1 was observed with BCG/placebo, while with BCG/Canvaxin™ of 34.9 months, showing that BCG shows the primary responsibility for the increased survival rate.

Recently, Nishida et al., (2019) [60] conducted a study administering BCG-CWS as an immune adjuvant, along with tumor-associated antigen-specific peptides (TAA) in 18 patients with advanced solid cancer, 7 of whom had MM. In this work, the effects of BCG-CWS on the immune system of patients with advanced cancer, mainly adaptive cellular immunity and the immunological phenotypes of T-cell subsets, were evaluated. An increase in neutrophils and monocytes was observed as a function of the time suspected to be related to BCG-CWS doses (50 mg, 100 mg, and 200 mg). Furthermore, there was an induction and differentiation of CD4^+^ T cells through direct activation of innate immunity. It is important to note that BCG products have immunomodulatory effects because they contain several unique bioactive components capable of influencing the immune system through various types of pattern recognition receptors (PAMPs) [95,96]. In this way, the maturation and differentiation of DCs occur in professional APCs, which indirectly activates the functional differentiation of antigen-specific CD4^+^ T cells and stimulates the antigen’s cross-presentation to CD8^+^ T cells [97]. Therefore, based on these results, we can observe a promising role of BCG in cancer.

BCG induces type I IFN and results in signaling and APC activation. This inflammatory axis could also restrain tumor growth. It was shown, in melanoma patients, that type I IFN is a transcriptional signature associated with tumor-infiltrating T cells, playing a crucial role in tumor-initiated T cell priming–infiltrating T cells [98,99]. The binding of cyclic dinucleotides (CDNs) also results in an inflammatory cascade involving TBK1 activation, IRF-3 phosphorylation, and type I IFN and other cytokines [30,32].

Therefore, BCG’s mycobacteria could act as a non-specific adjuvant, improving T-cell response against melanoma in association with other microorganisms that may impact antitumor immunity in human cancer patients, as discussed above [86,88,100,101,102]. In another study using melanoma patient’s materials, T cell clones recognizing naturally processed cancer antigens that are cross-reactive with microbial peptides were found. This cross-reactivity was shared between tumor MHC class I-restricted antigens and an enterococcal bacteriophage [90]. Mycobacteria also share antigens with human tissue, accounting, in part, for the production of autoantibodies in mycobacterial infections. Mouse monoclonal anti-*M. bovis* antibodies were found to recognize autoantigens, such as thyroglobulin, myosin, actin, and collagen [101]. It is important to note that BCG therapy can induce systemic autoimmune phenomena, particularly arthritis [103]. The mechanism of action of BCG-induced arthritis is not entirely known. The most likely explanation is based on experimental studies of adjuvant arthritis through a molecular mimicry [104]. In BCG-treated bladder cancer patients, urothelial cells had strong MHC class-2 antigens (HLA-DR) expression, which persisted several months after therapy, suggesting continuous CD4 T cell activation by antigens shared between mycobacterium and cartilage proteoglycan could be responsible for autoimmune arthritis [102]. Therefore, a cautionary note should be made, considering the potential risk of long-term autoimmune disease development, especially when genetically modified BCG strains are used.

## 6. BCG’s Trained Immunity in Cancer

Trained immunity is related to innate immune cells that develop durable, functional modifications similar to immunological memory. Epigenetic changes and cellular metabolism cause these modifications. For trathis to occur, it is necessary to mobilize gene sequences with regulatory elements [105]. Training immunity is needed to mobilize myeloid cells after BCG [89], beta-glucans [106], and bacterial lipopolysaccharide (LPS) stimulation [107,108]. How innate cells develop long-lived innate immune responses with memory is not entirely understood. It is known that long-lived memory in hematopoietic cells that produce trained immunity involves genetic reprogramming and autonomic plasticity [108,109]. The innate immune response’s memory can be induced by epigenetic factors, regulation of transcription factors, and genetic reprogramming to be long-lived in the circulatory system [110]. Kaufmann et al., (2018) demonstrated that BCG also induced long-term changes in myeloid cell-progenitors through trained immunity [105].

Like TLRs, BCG could modify progenitor cell populations with new open chromatin regions that persist [110]. It has been described that the cells of the innate response develop genetic reprogramming dependent on an enhancer-binding protein β (C/EBPβ) induced by LPS and Gram-negative bacteria [108]. Thus, short-term cell signaling would cause gene-specific chromatin modifications dependent on C/EBPβ, leading to the induction of hematopoietic cells with trained immunity and immune memory. In hematopoietic cells from C/EBPβ-deficient mice, the innate immune memory response and epigenetic changes are not evident. However, the cell phenotype and gene expression profile are indistinguishable from untreated wild-type control animals [108].

Innate immune cells, such as macrophages, monocytes, and NK cells, can develop long-term reprogramming by activating danger recognition receptors (DAMPs), PAMPs, and TLRs involving epigenetic mechanisms [111]. Similar to a classical immunological memory, NK cells undergo a secondary expansion after restimulation, degranulate faster, and produce cytokines inducing a protective immune response [112]. It is also proposed that NK prime monocytes in the bone marrow, inducing long-term immune responses contributing to innate immunity [113]. According to Netea et al. (2016) [110], the innate immune cells with enhanced epigenetic status present persistence of histone marks determining latent enhancers-like, resulting in more robust activation in response to restimulation.

The fact that the epigenetic modifications of innate immunity cells persist is indicative that this functional state can be somewhat trained to the immediate increase in pro-inflammatory response, probably soon after a second contact with similar molecules during BCG primming. The regulation of central cell metabolism acts on epigenetic enzymes produced from transcription factors in the nucleus. After the initial challenge with live pathogens or vaccines, the intracellular signaling pathways increase the pro-inflammatory response by increasing certain transcription factors. These factors return to the baseline after priming but can lead to a transient increase in myelopoiesis to respond to other stimuli [114]. Therefore, induction of trained immunity can provide non-specific effects that can promote the innate immune response [115].

Another essential factor during trained myeloid cell immunity is PU.1 (a transcription factor enconded by the SF1 gene). Hematopoietic transcription factor PU.1 can act as an antagonist of GATA-1 (erythroid transcription factor) [116,117,118], thus composing an irreversible mechanism for the erythro-myeloid lineage commitment [116,119,120]. During their development in the thymic microenvironment, cells become single positive for CD4 or CD8 T lymphocytes, regulatory T (Tregs), or invariant NKT cells (iNKT). This mechanism appears to characterize various T cell ontogeny stages [121], and PU.1 gene transcription is off before intrinsic cell commitment to the T-cell lineage when lymphocytes arrive at the thymus microenvironment [120]. Again, this could be evidence that the trained immune response may regulate the adaptive immune response beneficially in the antitumor response [121,122]. It has also been described that in humans, BCG vaccination readily promoted the production of IL-1β, TNF-α, and IL-6 by monocytes upon ex vivo stimulation with unrelated pathogens and upon ex vivo stimulation with unrelated pathogens [123,124]. BCG also induced human monocyte epigenetic reprogramming NOD2-dependent modifications [123]. In this case, cross-protection was supposedly shared by various pathogens [125] since BCG, a bacterial vaccine, also protected against an attenuated viral strain of Yellow Fever [126] or viral respiratory diseases [127]. Studies have also shown that long-term cross pathogen effects (both in vaccination and infection or in transplantation and in vitro differentiation experiments) suggest that immune memory’s epigenetic mechanisms also occurred in long-lived cells [105,110].

Immune response in the elderly is deficient mainly on adaptive responses, but the innate immune response function is supposedly intact [127,128]. In this case, BCG could be a tool to increase the pro-inflammatory response by inducing an anti-neoplastic response in elderly individuals. Therefore, it is noteworthy that in the specific case of using BCG in the case of immunotherapy for cancer, further studies should depend on the particular case of neoplasia and where it develops. Furthermore, having innate immunity and in a trained manner responding promptly with the BCG stimulus can lead to the modulation of the adaptive immune response so that tumors can be eliminated. Further studies must be conducted to certify the use of BCG in different situations [129]. BCG’s beneficial effects need to be evaluated in terms of the dose and frequency of the vaccine’s challenge, tumor environment and capacity of the memory, and innate immune cells to shape the adaptative immune responses. These studies can be an enriching approach to future translational and clinical studies.

## 7. Conclusions

Finally, the use of BCG has shown promise for the treatment of malignancies, such as bladder cancer [130], melanoma [131], leukemia [132], and lymphoma [133]. In all these studies, BCG heterologous effects are dependent on innate immunity, probably involving trained immunity. Although the direct impact of the pro-inflammatory response may be linked to the reaction against the tumor, the memory of innate memory cells could also be involved in the long-term responses in this type of treatment. Especially in bladder cancer, BCG’s effects against tumor cells seem to be essentially dependent on trained immunity since patients treated with BCG who were unable to mount a balanced innate immune response presented lower survival [134]. Trained immunity mechanisms induced by BCG could also protect against viral infections such as Yellow Fever or even against Covid-19 [115,126]. Therefore, BCG vaccination appears to be highly relevant with non-specific benefits and effects, especially if combined with other available therapies as discussed above.

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
