# Peer review of "Bacillus Calmette–Guérin Immunotherapy for Cancer"

_vaccines, 2021, doi:10.3390/vaccines9050439_

Round 1

Reviewer 1 Report

Authors aim to address several interventions using BCG on neoplasms such as Leukemia, Bladder Cancer, Lung Cancer, and Melanoma, describing treatments and anti-tumor responses promoted by BCG. Additionally, in this review, there is a deepening on the effective response during cell-target therapy when combined with the anti-PD-1 antibody, which stimulates T-cell responses against melanoma. The review is well structured. After carefully reading, this manuscript lacks totally to review the effects of BCG treatment in the immune response of infiltrating gd T cells in Bladder cancer. Moreover, the Authors should add references in lines 228-231. In some points, the review appears redundant, authors should check it. There are some errors in formatting.

Author Response

Dear Sir, Thanks for carefully revise our manuscript.

Reviewer 1

Point 1- “After carefully reading, this manuscript lacks totally to review the effects of BCG treatment in the immune response of infiltrating gd T cells in Bladder cancer.” Asnwer: Text and REF 41 added to lines 115-119: “A recent study showed that patients with NMIBC revealed an impressive improvement in BCG response when immunized percutaneously before intravesical. gd T and NK cells' weak cytotoxic responses were reported and associated with poor local BCG responses, a situation reversed by previous systemic immunization [41].”

Point 2-“Authors should add references in lines 228-231.” Answer: References added into the text, which now is contained in lines 229-235.

Point 3-“In some points, the review appears redundant. There are some errors in formatting”. Answer: Many redundant parts were suppressed or reallocated. These are listed in a cover letter to the editor, as below. English spelling was corrected. 

Finally, as requested the cover letter contains: 

"Now, after carefully answering the referee's questions and suggestions, the manuscript was modified as follows:

1-Suppressed or modified texts from the first submission: lines 73-92; 93-107, 108-114 lines in the previous version.

2-Modified words and sentences: a)Words : “non-specific” (line 45), Non-invasive (90), standard therapy (113), “than, globally, Brazilian” (lines 129-130), “proved to be” (line 74); “Group (SWOG) and the Memorial Sloan-Kettering” (lines 76-77); “Studies to evaluate other BCG strains,” line 78); meaning 40% (line 229), Lowering this vaccine's amount (235-236), different (265), also (295), related to (line 299), crucial (360), italic M. bovis and M. tuberculosis (lines 41 and 372, respectively), ref, 110 (line 388), highly (478). b) Sentences: “environmental exposure to tobacco smoke increases urothelial developing BC's risk.” (lines 55-56); ... note should be made, considering the potential risk of long-term-autoimmune disease development (lines 479-480).

3-New reallocated or modified paragraphs in the revised form: text lines 98-101; 84-89; 91-95; 102-112; 115-119; 304-324; 369-381."

PS- Authors and emails*: Please, notice that all the institutional emails were also added to the revised form (maiara.bonfim@ufba.br; perielavasconcelos@id.uff.br; jomengel@ioc.fiocruz.br), and Ms "Vasconcelos de Souza" is now corrected in the present version since her name was submitted correctly, but it was in another order in the former text."

Reviewer 2 Report

BCG, a vaccine against tuberculosis, is also a treatment against bladder cancer. Can it also be a treatment for other types of cancer? BCG has indeed been found to increase the activity of the immune system non-specifically in a variety of contexts and cancer types. The authors explore and describe that possibility in this concise review.

In this article, the authors start by a brief historical overview of BCG in the field of cancer therapy. It then moves onto BCG specifically for bladder cancer as a “gold standard” and possible mechanism of action in that type of cancr, as well as leukemia, lung cancer, melanoma

  1. Bladder Cancer

Define/explain Ta, Tis, and T1

Line 86/87: “Southwestern Oncology Group (SWOG) 86 from the USA and the Memorial Sloan-Kettering” -> Memorial is also in the USA

Line 96: “BCG’s Anti-Tumor effect, its mechanism of action has not yet been fully elucidated” -> the authors discuss the non-specific immune effects of BCG. Could there also be a specific immune effect? By molecular mimicry, BCG could elicit an immune response against a specific peptide, possibly a neoepitope, found in some cancers, such as bladder cancer. There’s work showing that Bacteroides triggers myocarditis through a myosin mimetic peptide, efforts mapping epitopes present in food that match autoimmune epitopes more generally, and, perhaps most relevantly, the description of an enterococcal peptide that mimics a melanoma peptide, explaining why presence of enterocci led to better melanoma immunotherapy.

https://science.sciencemag.org/content/366/6467/881.abstract

https://academic.oup.com/intimm/article/32/12/771/5893934

https://science.sciencemag.org/content/369/6506/936.editor-summary

It would be worth discussing these studies in the context of BCG and cancer

  1. Melanoma

The authors mention a study showing how the microbiome impacts the response to immune checkpoint blockade and provide one highly cited reference, more should be provided as now mutiple independent labs have been building on and expanding these findings

A molecule that is not mentioned at all and shouldn’t is STING, as it “warms up” the tumor for T cell attack, very reminiscent of what BCG might be doing, maybe there’s a link there. Example:

https://www.cell.com/cell-reports/fulltext/S2211-1247(15)00432-5

The whole review only has positive aspects of BCG in bladder cancer. It is utmost important to mention potential shortcomings as well: not just lack of specificity but also potential complications from using it, e.g. autoimmunity in very few cases

https://pubmed.ncbi.nlm.nih.gov/11334488/

A more general comment, but the review would benefit from having at least one figure/schematic.

Author Response

Dear Sir,

Thanks for carefully revise our manuscript.

Reviewer 2

Point 1-... “BCG specifically for bladder cancer as a “gold standard”. Corrected to: “standard therapy”.

Point 2-“Bladder Cancer: Define/explain Ta, Tis, and T1”. These are explained now, into lines 92-93.

Point 3-Line 86/87: “Memorial is also in the USA” Corrected now, in lines 77-78.

Point 4-Line 96: “BCG’s Anti-Tumor effect, its mechanism of action has not yet been fully elucidated” -> the authors discuss the non-specific immune effects of BCG. Could there also be a specific immune effect? Added REFs 27-28 explaining specific T cells, text into the lanes 85-90.

Point 5-Comments:”…molecular mimicry: BCG could elicit an immune response against a specific peptide, possibly a neoepitope, found in some cancers, such as bladder cancer”. “Bacteroides … an enterococcal peptide that mimics a melanoma peptide, explaining …enterococci led to better melanoma immunotherapy”. Added refs. 365 and 382, with a new paragraph explaining this mechanism (lines 370-382). Studies were discussed in the context of cancer.

Point 6- “Melanoma: The authors mention a study showing how the microbiome impacts the response to immune checkpoint blockade and provide one highly cited reference”. REF 5 was added to the 91 REF, and other REFs as 92, 93, 94, 95, 97, 97 are listed and explained in lines 331-335.

Point 7- “A molecule that is not mentioned at all and shouldn’t is STING”. STING mechanism is detailed in REFs 34, 35, 36, 37, 38, 39, 40. A new text was added explaining it, in lines 113-123.

Point 8- Comments: …potential complications from using it, e.g. autoimmunity in very few cases. Autoimmunity-related disorders or complications are now explained in lines 370-382 with new REFs 95, 106, 107 108, 109.

Point 9- Figure/scheme. We have found that explain BCG acting as a general scheme to cancer is deeply complex. Finally, a figure was not added. We suggest that BCG appears to have a different action in the microenvironment of each different tumor.

Finally, as requested the cover letter contains: 

"Now, after carefully answering the referee's questions and suggestions, the manuscript was modified as follows:

1-Suppressed or modified texts from the first submission: lines 73-92; 93-107, 108-114 lines in the previous version.

2-Modified words and sentences: a)Words : “non-specific” (line 45), Non-invasive (90), standard therapy (113), “than, globally, Brazilian” (lines 129-130), “proved to be” (line 74); “Group (SWOG) and the Memorial Sloan-Kettering” (lines 76-77); “Studies to evaluate other BCG strains,” line 78); meaning 40% (line 229), Lowering this vaccine's amount (235-236), different (265), also (295), related to (line 299), crucial (360), italic M. bovis and M. tuberculosis (lines 41 and 372, respectively), ref, 110 (line 388), highly (478). b) Sentences: “environmental exposure to tobacco smoke increases urothelial developing BC's risk.” (lines 55-56); ... note should be made, considering the potential risk of long-term-autoimmune disease development (lines 479-480).

3-New reallocated or modified paragraphs in the revised form: text lines 98-101; 84-89; 91-95; 102-112; 115-119; 304-324; 369-381."

PS- Authors and emails*: Please, notice that all the institutional emails were also added to the revised form (maiara.bonfim@ufba.br; perielavasconcelos@id.uff.br; jomengel@ioc.fiocruz.br), and Ms "Vasconcelos de Souza" is now corrected in the present version since her name was submitted correctly, but it was in another order in the former text.

Round 2

Reviewer 2 Report

Line 312 – “Besides” is repeated – typo

Line 327 - “New strategies that enhance therapies against metastatic melanoma (MM) have been 327 studied, such as BCG, despite current knowledge” – this sentence doesn’t make sense, please rewrite

The authors now corrected parts of the text and added paragraphs on STING and molecular mimicry

Make sure to proofread text again